# Personality and Family Risk Factors for Poor Mental Well-Being

**DOI:** 10.3390/ijerph20010839

**Published:** 2023-01-02

**Authors:** Maya Peleg, Ora Peleg

**Affiliations:** 1Social and Organizational Psychology, Bar Ilan University, Ramat Gan 5290002, Israel; 2Education and School Counseling Departments, Max Stern Yezreel Valley College, Yezreel Valley 1930600, Israel

**Keywords:** differentiation of self, anxiety, self-regulation, mental well-being

## Abstract

There is evidence that differentiation of self (DoS) contributes to the regulation of emotions at a young age, resulting in reduced anxiety and improved mental well-being. However, there is little evidence of the relationships between these four dimensions (DoS, self-regulation, anxiety, and mental well-being), or of the potential mediating role of self-regulation and anxiety. Our primary goal was therefore to consider the links between DoS, self-regulation, anxiety, and mental well-being. It was hypothesized that DoS (emotional reactivity, I-position, emotional cutoff, fusion with others) will be positively associated with mental well-being through the mediation of self-regulation (promotion-focused, prevention-focused) and anxiety. The study included 460 participants with a mean age of 41.18 (*SD* = 14.97, range = 19–60). Of them, 224 (48.7%) were women. Participants filled out four questionnaires: the Differentiation of Self Inventory–Revised, General Regulatory Focus Measure, the anxiety scale from DASS–21, and the Warwick–Edinburgh Mental Well-being Scale. The findings showed that emotional cutoff positively predicted prevention and anxiety, but did not predict promotion. In addition, promotion positively predicted mental well-being. Finally, promotion mediated the relationship between I-position and mental well-being. The results indicate that well-differentiated individuals function optimally and enjoy good quality of life.

## 1. Introduction

According to Bowen’s family systems theory, the emotional, value, belief, and behavioral patterns in a system (i.e., family) are passed down from generation to generation [1]). Bowen also argued that childhood and adolescence are critical periods for achieving developmental tasks, including autonomy and emotion regulation. With this in mind, we explored the family and personality risk factors that may harm mental well-being in adulthood. Specifically, we examined whether and how individuals who are well differentiated are able to regulate their emotions and reduce anxiety and, as a result, enjoy greater mental well-being.

## 2. Differentiation of Self

In his family systems theory, Bowen [1] claimed that children and adolescents learn to be less dependent on their parents and to develop symmetrical relationships with them, increasing their personal ability to make decisions independently and to deal with conflicts and stressful situations in an emotionally balanced manner. He also argued that it is possible for the individual to protect his or her mental well-being and maintain functionality by breaking free of symbiotic ties in the family of origin that may lead to pathologies [1,2,3]. He suggested that one of the central family patterns contributing to this process is differentiation of self (DoS), which enhances the quality of life and mental and physical health of family members, affecting their perceptions of stressful events and their thought processes. 

DoS is defined at two levels. At the intrapersonal level, it describes the ability to maintain a balance between intellectual and emotional functioning—namely, to be able to consider effective coping strategies in stressful situations without being emotionally overwhelmed. At the interpersonal level, it taps the degree of intimacy and autonomy in significant relationships. People who are poorly differentiated find it difficult to reduce their anxiety and regulate their emotions, which hinders their capacity to reflect on reality and assess it in a more balanced way, to deal more adequately with life stressors, and to tolerate ambiguity and uncertainty [1].

Theoretically, DoS includes four dimensions. Emotional reactivity describes the tendency to react to stressful stimuli automatically and emotionally, and the inability to remain calm in the presence of significant others in stressful situations. I-position taps individuals’ ability to adhere to their needs, desires, and thoughts, even when these do not coincide with the opinions of close others. Emotional cutoff describes individuals’ tendency to isolate themselves physically, emotionally, and verbally; to avoid sharing feelings; and to rigidly disconnect from significant others as a way of managing stress and intimate relationships. Finally, fusion with others reflects the tendency to form symbiotic relationships and to seek the approval and acceptance of others. In general terms, DoS can be described as the individual’s ability to define the self without being emotionally overwhelmed and to maintain intense and stable relationships [3]. Accordingly, well-differentiated people are able to develop, invest energy in life’s tasks, deal effectively with stressful events, and develop an awareness of their emotions that increases their ability to regulate emotions [4]. In contrast, less differentiated individuals have distinctly intense anxiety because they have trouble developing emotional separation.

Empirically, low DoS has been found to reinforce the false self, increase anxiety [5], correlate with physiological symptoms [6], and impact physical health [6,7]. It has also been shown to hinder the individual’s ability to deal positively with important tasks and stressful events and to regulate emotions [8,9].

## 3. Self-Regulation

Higgins’ [10,11] self-regulation theory expands upon the basic hedonist motivational principle in social psychology, according to which people are motivated to feel pleasure and enjoyment and avoid pain. According to Higgins, two separate and independent orientations underlie people’s motivation to adapt themselves, their behaviors, and their perceptions to desired goals and standards: promotion-focused regulation and prevention-focused regulation (see also [12,13]).

People who are characterized by promotion-focused regulation strive to fulfill their goals while maintaining their values. They tend to take risks, to adopt strategies of actively searching for innovative ideas, and to improve their performance after failure or after coping with a difficult task (Higgins, [10,11]) Such people have been reported to experience higher levels of well-being [14,15] quality of life [16], mental well-being in the workplace and in their private lives [17,18] and positive emotions [19]. Moreover, they were less likely to experience negative emotions when dealing with conflicts (e.g., in marital relationships) as well as with failures [20].

In contrast, individuals characterized by prevention-focused regulation are motivated to act for desirable or proper self-fulfillment through responsible and cautious strategies [10,11]. They act as expected of them, striving to achieve their goals while maintaining safety and security, examining potential losses, and avoiding risk, failure, and pain. Such people are characterized by a conservative style of thinking in which they carefully weigh alternatives, take few risks, and prefer a strategy of repetition over innovation. They experience success or failure as the presence or absence of adverse outcomes and typically do not try to tackle failed tasks again [13,21,22]. People with prevention-focused regulation have reported lower levels of subjective well-being, vitality, self-control, satisfaction with life [23,24,25], and quality of life [16], as well as higher levels of somatic symptoms [24]. They experience a negative psychological process aimed at preventing a negative outcome and avoiding pain rather than increasing joy, which in turn may increase emotional distress and anxiety [25].

Self-regulation has been found to mediate the relationship between self-generated stress and psychological well-being [26]. Additionally, the ability to regulate emotions (promotion-focused regulation) has been found to have positive effects on an individual’s internal dynamics and social relationships, whereas difficulties in emotion regulation (prevention-focused regulation) have been associated with various psychological struggles [27,28]. In addition, poor orientation to self-regulation has been found to be linked to aggressiveness, antisocial behaviors [29], depression, and heightened anxiety [28].

## 4. Anxiety

Anxiety is defined as the individual’s response to a real or imagined threat [3]. According to the DSM-5 [30], anxiety is a fear unrelated to a real external threat or an over-interpretation of an existing situation, which is usually irrational and subjective. It has been linked to physiological and psychological responses (e.g., muscle tension, sleep disturbances, rapid fatigue, restlessness, irritability, and difficulty concentrating) that appear following anticipation of a threat and lead to cautious or avoidant behaviors [30,31]. Several studies have shown that increased anxiety is associated with lower DoS [32,33], poor self-regulation, and poor mental well-being [34,35].

## 5. Mental Well-Being

Well-being represents the psychological component of the term “quality of life” [36] Describing positive aspects of human experience [37] this concept encompasses a range of emotions and experiences, such as thriving, vitality, satisfaction, and health. Mental well-being enables the development of the individual’s personal, social, and occupational potential [38,39].

There is evidence that poor mental well-being is associated with negative feelings, such as psychological stress, low self-esteem, loneliness, and depression [40]. Studies of the factors contributing to mental well-being found it to be affected by self-regulation [41] and DoS [33], and to be negatively associated with anxiety [41].

## 6. Goals and Hypotheses

In sum, there is evidence that DoS contributes to the regulation of emotions, resulting in reduced anxiety and improved mental well-being. Moreover, low DoS, poor self- regulation, and a high level of anxiety have been found to be inversely associated with mental well-being. However, there is little evidence of the relationships between these four dimensions (DoS, self-regulation, anxiety, and mental well-being) or of the potential mediating role of self-regulation and anxiety in this context. In light of recent studies showing that mental well-being may contribute significantly to mental and physical health [42], it is important to investigate the risk factors that increase predisposition to poor mental well-being, to understand the psychological mechanism that activates it, and thus to help improve it. It is therefore reasonable to assume that difficulties in self-regulation and increased anxiety may predispose individuals to poor mental well-being and play an important role in mediating the relationship between DoS and mental well-being.

In the current study, we explore and expand upon family systems theory [3] to include mental well-being. Our primary goal was therefore to consider the links between DoS, self-regulation, anxiety, and mental well-being. The examination of this model may help deepen understanding of the factors that directly or indirectly contribute to mental well-being and thus help improve quality of life. Thus, we assumed that susceptibility to poor mental well-being is influenced not only by one’s level of self-regulation and anxiety, but also by the extent to which one’s self-regulation and anxiety are dependent on DoS. Hence, our main hypothesis is:DoS (emotional reactivity, I-position, emotional cutoff, fusion with others) will be positively associated with mental well-being through the mediation of self-regulation (promotion-focused, prevention-focused) and anxiety.

The following sub-hypotheses are derived from this hypothesis:a.DoS will have an inverse association with prevention and anxiety, and a positive association with promotion and mental well-being.b.Anxiety will be positively associated with prevention, and negatively associated with promotion and mental well-being.c.Mental well-being will be positively associated with promotion, and negatively associated with prevention.

## 7. Method

### 7.1. Participants

The study included 460 participants with a mean age of 41.18 (*SD* = 14.97, range = 19–60). Of them, 224 (48.7%) were women, 246 (53.5%) were college educated, 109 (23.7%) were single, 299 (65.0%) were married, and the rest had another marital status (11.3%). Participants were recruited from all regions of the country by systematic sampling. Inclusion criteria were individuals whose parents live together, who have good reading comprehension, and who do not suffer from reading or writing disabilities.

### 7.2. Instruments

**DoS** was measured by the **Differentiation of Self Inventory–Revised** (DSI–R), ref. [43,44], which was validated and translated to Hebrew [45]. The DSI–R includes 46 items divided into four subscales: emotional reactivity, I-position, emotional cutoff, and fusion with others (sample item for emotional reactivity: “People have remarked that I’m overly emotional”). Participants answered each item on a Likert scale of 1 (not at all like me) to 6 (very much like me). Subscale scores were calculated by averaging the mean scores. Greater DoS is indicated by lower means for emotional reactivity, emotional cutoff, and fusion with others, and by higher means for I-position. Cronbach’s alpha coefficient indicated good reliability for the emotional reactivity, I-position, and emotional cutoff subscales (0.86, 0.80, 0.83, respectively), and acceptable reliability for the fusion with others subscale (0.73).

**Self-regulation** was assessed by the **General Regulatory Focus Measure** (GRFM) [46], translated to Hebrew and adapted for the purposes of the present study. This 18-item questionnaire includes two measures. Half the items represent promotion-focused regulation, i.e., pursuing desirable outcomes and viewing goals as ideals, indicating good self-regulation (sample item: “I frequently imagine how I will achieve my hopes and aspirations”). The other nine items represent prevention-focused self-regulation, i.e., avoiding undesirable outcomes, indicating poor self-regulation (sample item: “In general, I am focused on preventing negative events in my life”). Responses are rated on a Likert scale of 1 (very untrue for me) to 9 (very true for me). Scores are calculated by averaging item scores for each of the two scales. Cronbach’s alpha coefficient indicated good reliability for both subscales (0.88 for promotion and 0.83 for prevention).

**Anxiety** was measured by the anxiety scale within the **Depression Anxiety Stress Scales** (DASS–21) [47], translated to Hebrew and adapted for the purposes of the current study. The anxiety scale includes seven items out of the original 21, which also measure depression and stress. A sample item for anxiety is: “I had a feeling of shakiness.” Responses are rated on a Likert scale (0 = does not characterize me at all, 3 = characterizes me most of the time). Scores are calculated by averaging the item scores in the scale. Cronbach’s alpha coefficient indicated good reliability for the questionnaire (0.93).

**Mental well-being** was examined by the **Warwick–Edinburgh Mental Well-being Scale** (WEMWBS) [48], translated to Hebrew and adapted for the purpose of the present study. Items cover various aspects of eudemonic and hedonic mental well-being and are worded positively. Sample item: “I have been feeling optimistic about the future.” In this 14-item scale, participants respond along a 1–5 Likert scale. Item scores were averaged to produce a total score ranging from 1 to 5, with higher scores representing higher levels of mental well-being. Cronbach’s alpha coefficient indicated good reliability for the questionnaire (0.93).

### 7.3. Procedure

After receiving ethics approval from a college in the North of Israel (202311 YVC EMEK), questionnaires were distributed by a survey company. Participants were recruited by a systematic sample from all regions of Israel. All participants signed an informed consent form (the first page of the questionnaire) and answered the questions online. Completion of the questionnaires was voluntary. Participants were promised anonymity and discretion and were informed that they could stop filling out the questionnaires at any time.

### 7.4. Data Analysis

Data analysis was conducted using SPSS V26 software. The final mediation model was yielded by AMOS software. As a preliminary analysis, descriptive statistics were calculated for the study variables.

The research hypotheses were tested in three stages. In the first stage, we ran correlations between the variables. In the second stage, we ran multiple regressions predicting well-being, with gender, age, education, and marital status as covariates. In the third stage, AMOS was used to calculate a path analysis model. The demographic variables mentioned above were added to the model according to the modification indices.

## 8. Results

### 8.1. Preliminary Analyses

Table 1 presents means, standard deviations, and the distribution of the study variables. The distribution indices (skewness and kurtosis) indicate the variables were distributed approximately normally, allowing for parametric analysis. As the study variables were significantly related to only two DoS indices—I-position and emotional cutoff—only they were included in the analyses (and in all tables).

Table 2 displays correlations between DoS, self-regulation, anxiety, and mental well-being. I-position was positively associated with promotion and mental well-being, and negatively associated with anxiety, prevention, and emotional cutoff. Emotional cutoff was negatively correlated with promotion and mental well-being, and positively correlated with prevention and anxiety. Anxiety was negatively related to promotion and mental well-being, and positively related to emotional cutoff and prevention. Finally, mental well-being was positively associated with promotion and negatively associated with prevention.

Table 3 displays multiple regression analyses, with well-being as the dependent variable, sociodemographic variables at a first step as control variables, and the variables of the study (DoS, self-regulation, and anxiety) as independent variables at the second step. As can be seen from the table, when controlling sociodemographic variables, both DoS variables and promotion make a significant unique contribution to the regression model whereas prevention and anxiety make a non-significant unique contribution.

### 8.2. Mediation Model

The correlations presented above show that I position (predictor variable X_1_) and emotional cutoff (predictor variable X_2_) were correlated with mental well-being (outcome variable), as well as with prevention, promotion, and anxiety (the three mediating variables). In addition, all three mediators were found to be correlated with mental well-being (outcome variable).

Results of the mediation model appear in Figure 1 and Table 4. Gender, age, education, and marital status were included as covariate variables (the last variable was entered as a dummy). As expected, I-position positively predicted promotion and negatively predicted prevention and anxiety. In addition, I-position positively and directly predicted well-being. Emotional cutoff positively predicted prevention and anxiety, but not promotion. In addition, emotional cutoff negatively and directly predicted mental well-being. Finally, promotion positively predicted well-being. Contrary to the research hypothesis, promotion and anxiety were not significant predictors of mental well-being.

Promotion mediated the relationship between I-position and mental well-being, while anxiety and prevention did not mediate this relationship. None of the three (promotion, prevention, or anxiety) mediated the relationship between emotional cutoff and mental well-being.

## 9. Discussion

The current study explored personality and family risk factors for poor mental well-being. Specifically, we considered the role of self-regulation and anxiety in the relationship between DoS and mental well-being. Our overall results demonstrated that two dimensions of DoS were associated with mental well-being: I-position was associated with mental well-being both directly and through the mediation of promotion (self-regulation), whereas emotional cutoff was only associated with mental well-being directly.

The main result is that promotion-focused self-regulation mediated the relationship between I-position and mental well-being. This partially supports Hypothesis 1 and study findings indicating that the more individuals stick to their own goals and desires, and perceive themselves as free to fulfill these goals, the more likely they are to experience positive emotions and happiness [49]. I-position taps the ability to openly and directly express needs and desires, and to focus on personal goals and objectives without being distracted by emotional fluctuations. We suggest that individuals whose decisions and choices do not depend on others, and who adhere to their own desires and needs, will feel satisfied and have a strong sense of control over their lives. This may help them manage and regulate their emotions, which in turn may increase their mental well-being.

It is important to note that, in addition to the significant mediation model that was revealed, I-position directly predicted mental well-being (Hypothesis a). This may be explained in terms of person-centered theory [50,51,52], which stipulates that the real self represents one’s conceptions of how one is, whereas the ideal self represents one’s perceptions of how one would most like to be. When expectations of oneself are incongruent with and contradictory to the real self, they may become threatening and result in a high level of anxiety and frustration. Viewed through the lens of person-centered theory, then, it is plausible that there is congruency between the real and ideal self among individuals who stick to their desires and expectations, increasing their satisfaction with life.

An interesting finding is that self-regulation and anxiety did not mediate the relationship between emotional cutoff and mental well-being. However, emotional cutoff was found to be positively related to prevention, anxiety, and mental well-being (Hypothesis a). This is in line with studies showing that low DoS, and specifically high emotional cutoff, can increase anxiety to the detriment of mental well-being [33,49]. We suggest that people who do not express their feelings and desires directly, but rather repress their feelings and tend to detach physically, emotionally, and verbally, may experience emotional relief in the short term, especially in symbiotic and intimate relationships. However, in the long term, they may experience higher levels of anxiety and emotional distress because by refraining from sharing their feelings, they may lack support from significant others. This pattern will make it difficult for them to effectively deal with stressful life events and enjoy good quality of life.

Only two dimensions of DOS, namely, I-position and emotional-cutoff, were found to predict anxiety, self-regulation, and mental well-being, partially supporting Hypothesis a and previous studies [7]. This is in keeping with a study that examined the contribution of DoS to satisfaction with life, which indicated that the dimension of I-position had the greatest influence on satisfaction with life, while emotional cutoff was the most detrimental [52]. As mentioned above, we suggest that the ability of individuals to create direct and clear communication, to stand up for themselves, and to meet their needs and feelings while receiving support from significant others may significantly improve their functioning and emotional state, which will ultimately improve their quality of life.

Another interesting and surprising finding is that anxiety and prevention did not predict mental well-being, partially refuting Hypotheses b, c and the results of previous studies [53]. This may be because people with high levels of anxiety use a variety of coping strategies. Thus, some use promotion and assertiveness, which increase their mental well-being and some use prevention and disconnection, decreasing mental well-being. Another possibility is that we examined chronic self-regulation, which describes the general emotional state of participants and not how they felt while filling out the questionnaires. In future research it is recommended to also include a state self-regulation questionnaire [54] in order to examine whether participants’ emotional state at the time they answer the questionnaires yields similar results.

Interestingly, women reported higher levels of promotion-focused self-regulation than men. This might be explained by gender differences from early childhood. For example, Weinberg and colleagues [55] reported that, beginning in infancy, girls are better able to regulate affect during the Still-Face experiment paradigm than boys. Some evidence suggests gender differences in the use of self-regulation strategies in toddler years, with boys more likely to seek distractions while girls prefer support from their mothers [56]. Girls also show better effortful control [57] than boys. In preschool years boys were found to have a higher prevalence of externalizing behaviors and physical aggression while girls were found to have decreased impulsivity [58]. As for empirical evidence in adulthood, one study [59] observed that men are more likely than women to disengage and avoid contact with people in uncomfortable situations. It is likely that these differences are due to physiological and hormonal differences, as well as to the different socialization process of boys and girls. Girls are taught to connect with significant others and talk openly about their feelings, while boys are expected to overcome their feelings of anxiety. Thus, there seems to be a general trend for girls to demonstrate better self-regulation than boys in early childhood, which probably persists throughout life.

Examining the relationship between the study variables and the background variables yielded a few interesting results. First, age was positively associated with I-position and mental well-being, and negatively associated with anxiety. This is consistent with literature suggesting that older age is associated with lower stress and depressive symptomatology, and with higher positive affect [60]. We suggest that the formation of self-identity and the extensive experience that has been gained during life in coping with various stressful events may decrease anxiety and improve the ability to deal with additional stressors and to use effective coping strategies, which in turn may lead to higher mental well-being. Second, having a college education was found to be negatively associated with anxiety, prevention, and emotional cutoff. This partially supports research indicating that education may improve self-regulation strategies [61,62]. Third, single people reported lower levels of I-position and mental well-being and a higher level of anxiety than married individuals and participants with a different marital status. It is likely that people who have experienced a long-term relationship may be more confident and reveal their feelings to a greater extent, which allows them to experience more positive feelings than single individuals, leading the latter to lower mental well-being. Additional support for this result can be seen in Soulsby and Bennett’s [63] study, which found that married people consistently report higher levels of psychological health than unmarried people, including widowed and divorced adults. In addition, it was reported that married people experience a lower number of days of poor mental health, a lower risk of depression, and higher life satisfaction [63]. Another possible reason for this result is in line with the previous explanation we proposed regarding the relationship between age and the research variables. It is suggested that most of the single participants in the present study were younger than their married counterparts and therefore might still be looking for meaning and direction in their lives. This stage of uncertainty about the future may be accompanied by ambivalence and low mental well-being.

### Limitations

The study is not without limitations. First, the online distribution of the questionnaires prevented access to potential participants who lack access to the Internet or have difficulties handling it. Second, the correlative nature of the study must be taken into account, as it does not allow to us to reach conclusions about cause-and-effect relationships between the study variables. Third, the use of self-report measures may lead to bias associated with social desirability. Fourth, DoS questionnaire data are based on personal reports of participants regarding their families. Information about family interactions is thus provided from the perspective of a single member, which may or may not reflect reality. Nonetheless, it is the individual’s point of view that determines how he/she feels and acts. To enrich our findings, future research should include the perspective of other family members, as well as external observers, to obtain more accurate information about family patterns. Finally, the current study focused on the concept of DoS, which is a key component of family systems therapy. It would be worthwhile for future studies to investigate additional concepts, such as intergenerational triangulation and intergenerational transmission.

## 10. Contributions

Notwithstanding the study’s limitations, results are consistent with theoretical assumptions that well-differentiated individuals function optimally and enjoy good quality of life [3]. Our research findings further support the family systems theory, providing empirical evidence to back Bowen’s assumption about the relationship between DoS, self-regulation, and mental well-being. In addition, they indicate that people who are well differentiated—i.e., manage to balance intimacy and autonomy, and know how to express their needs and desires without succumbing to the pressures of significant others—are able to regulate their emotions and consequently improve their mental well-being. The current study also contributes new data to the growing literature on the possible mediating role played by self-regulation in the relationship between the individual’s DoS and mental well-being.

## 11. Practical Implications

Our conclusions emphasize the considerable importance of family therapy for people wishing to improve their mental well-being. Bowen [1] posited that lower DoS should be seen as psychological dysfunction. In this sense, it is suggested that enhancing I-position and reducing emotional cutoff can improve self-regulation, which, in turn, is likely to improve mental well-being. Thus, we recommend that family therapists and counselors plan workshops and develop intervention techniques for both individuals and groups with the aim of increasing DoS.

## Figures and Tables

**Figure 1 ijerph-20-00839-f001:**
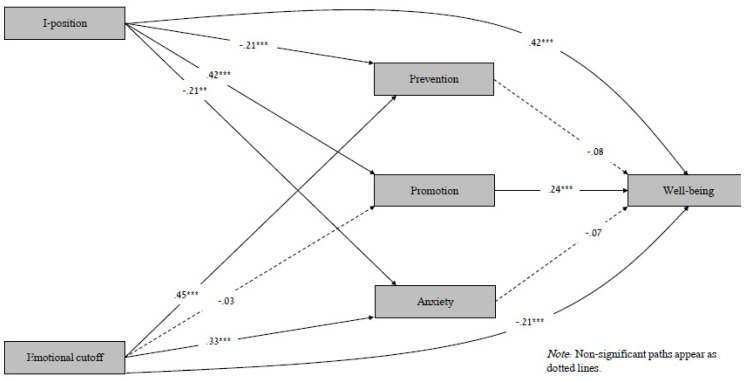
Summary of Mediation Model Using Path Analysis Model (N = 460). *** p* < 0.01. **** p* < 0.001.

**Table 1 ijerph-20-00839-t001:** Distribution and Descriptive Statistics of the Study Variables (N = 460).

	M	SD	Min.	Max.	Sk ^a^	K ^b^
**Differentiation of self**						
I-position	4.18	0.77	1.64	6.00	−0.25	−0.16
Emotional cutoff	2.87	0.86	1.00	5.25	0.14	−0.32
**Self-regulation**						
Prevention	4.86	1.78	1.00	9.00	−0.01	−0.56
Promotion	6.61	1.53	1.00	9.00	−0.59	0.22
**Anxiety**	1.72	0.77	1.00	4.00	1.17	0.59
**Mental well-being**	3.91	0.72	1.14	5.00	−0.72	0.66

^a^ Skewness. ^b^ Kurtosis.

**Table 2 ijerph-20-00839-t002:** Pearson Correlations between Study Variables (N = 460).

	1	2	3	4	5	6
**Differentiation of self**						
1. I-position	-					
2. Emotional cutoff	−0.26 **	-				
**Self-regulation**						
3. Prevention	−0.33 **	0.50 **	-			
4. Promotion	0.39 **	−0.14 **	0.17 **	-		
5. **Anxiety**	−0.33 **	0.38 **	0.40 **	−0.28 **	-	
6. **Mental well-being**	0.62 **	−0.41 **	−0.31 **	0.33 **	−0.34 **	-

** *p* < 0.01.

**Table 3 ijerph-20-00839-t003:** Regression Standardized Coefficient (Betas) and R-Squares of Multiple Regression Analyses for Predicting Well-Being (N = 460).

	Step I	Step II
**Age**	0.08	0.03
**Gender**	0.04	0.01
**Non-Academic Education**	0.01	0.08 *
**Family Status**		
Single	−0.08	−0.01
Married	0.04	0.08
**Differentiation of self**		
I-position		0.42 ***
Emotional cutoff		−0.20 ***
**Self-regulation**		
Prevention		−0.07
Promotion		0.25 ***
**Anxiety**		−0.07
$R^{2}$	0.031 *	0.516 ***
$\Delta R^{2}$	-	0.485 ***

** p* < 0.05. **** p* < 0.01.

**Table 4 ijerph-20-00839-t004:** Summary of Mediation Model (N = 460).

	Predicted Variables									
	Direct Effects								Indirect Effect	
	Prevention		Promotion		Anxiety		Mental Well-Being		Mental Well-Being	
Predictors	Beta	*p*	Beta	*p*	Beta	*p*	Beta	*p*	Beta	95% CI
**Demographic variables**										
Gender (Women)			0.08	0.030	0.09	0.019				
Age			−0.15	<0.001	−0.16	0.002				
Non-college educated			0.09	0.023	0.08	0.049	0.08	0.019		
Single					0.14	0.060	−0.09	0.007		
Married					0.12	0.054				
**Differentiation of self**										
I-position	−0.21	<0.001	0.42	<0.001	−0.21	<0.001	0.42	<0.001	0.13	0.07, 0.18
Emotional cutoff	0.45	<0.001	−0.03	0.462	0.33	<0.001	−0.21	<0.001	−0.06	−0.11, −0.02
**Self-regulation**										
Prevention							−0.08	0.079		
Promotion							0.24	<0.001		
**Anxiety**							−0.07	0.066		

## Data Availability

All the data is saved by the researchers in the Qualtrix application, but due to the assurance of confidentiality and discretion cannot be disclosed.

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
