# Peer review of "Personality and Family Risk Factors for Poor Mental Well-Being"

_ijerph, 2023, doi:10.3390/ijerph20010839_

Round 1

Reviewer 1 Report

The work entitled Personality and Family Risk Factors for Poor Mental Well-Being aim to examine the relationships between DoS, self-regulation, anxiety, and well-being, as well as to the mediating role of self-regulation and anxiety between DoS and well-being. I would like to congratulate the authors, since I think this is an interesting work, and has several strengths, as the background, or the mediating model which includes the different variables involved in the study. However, I think there are also some weaknesses that need to be considered and improved:

1.     I would recommend reviewing the writing of the manuscript, paying special attention to possible orthographic mistakes, and the review of the English writing.

2.     Abstract. I think it is not necessary to indicate that promotion predicts mental well-being, since previously it is written that it mediates the relationship between I-position and mental well-being, and this automatically implies that it predicts mental well-being. Perhaps, it is a good option to change the order of the findings. I think it is better, normally, to explain the relationship between variables first, then the prediction between them, and finally the mediation.

3.     Introduction, page 3. Self-regulation subsection. At the ending sentence, it is said that coping strategies are related to aggressiveness, antisocial behaviours… but although I understand that self-regulation has two dimensions which may are two different styles of dealing with their goals and values, I think it would be better to make this more explicit, that these self-regulation styles may be coping strategies, or perhaps, to indicate that a worse orientation to self-regulation would be linked to the variables explained (aggressiveness, antisocial behaviours…).

4.     Introduction, page 3. Regarding the last paragraph, which starts as “In sum…”, I think it would be better to include this in the next subsection about goals and hypotheses. Also in this paragraph, it is said that DoS is important at a young age. However, the study is not oriented to this goal. I was confused when I saw the sample, since according to this, I was expecting a young sample. I would recommend clarifying this.

5.     Methods. Instruments subsection. Cronbach’s alpha coefficients, or reliability coefficient in general, are about the scores of the test, and not about the test itself. Therefore, instead indicating that scale has a good reliability, authors should state that item scores showed a good reliability, or maybe better, just that Cronbach’s alpha coefficient indicated a good reliability.

6.     Methods. Data analysis subsection. Authors state that Pearson correlation analysis are conducted for continuous variables, but neither gender nor education are continuous variables. Additionally, I would recommend instead these analyses and ANOVA, a correlation analysis only between the variables of the study, and then a multiple regression analysis, including well-being as the dependent variable, sociodemographic variables at a first step as control variables, and finally the variables of the study (DoS, self-regulation, and anxiety) which were significant at correlation analysis. I think thit would facilitate the analysis and would provide significant information regarding the relationship of all the variables with well-being, controlling the others, in order to ascertain those variables which should be finally included in the mediation model.

7.     Results. Mediation model. I think the findings are not explained in the correct order. A mediation analysis is kind of a regression analysis, so you are conducting several regression analyses. First, I-position on prevention, I-position on promotion, I-position on anxiety, I-position on well-being. And second, prevention on well-being, promotion on well-being, and anxiety on well-being. Later the same for emotional cutoff. According to this, if it is previously indicated that a variable mediates between other two variables, this implies that the mediating variable predicts the outcome variable: “Promotion mediated the relationship between I-position and mental well-being” and later it is written “Finally, promotion positively predicted mental well-being”. I would recommend rewriting this subsection, perhaps indicating the relationships between variables first, and finally the ones which had a mediating role. In addition, according to the goals and hypotheses, I understood that anxiety was going to be tested as a mediator, not only between DoS and well-being, but also between self-regulation and well-being. I do not know if this was correct but there was no effect, or whether it never was tested, but I would recommend clarifying this and make the appropriate corrections. Finally, it is also written that “In addition, modification indices indicate a negative direct path between emotional cutoff and mental well-being”. Why is this not indicated for I-position as well? Both have a direct path to well-being, and it would be not necessary to specify this, since these variables are predictors, and well-being the outcome variable.

8.     Results. I recommend using the same names along all the manuscript. If the authors decided to use “prevention” in the diagram, and along all the manuscript to refer to one of the variables involved in the study, at table 5 this should be the same, instead using “avoidance”.

Author Response

Dear reviewer,

We are very grateful for your important comments, which undoubtedly contributed to the quality of the manuscript. Below are the corrections we have made:

Reviewer 1

  1. Comment: "I would recommend reviewing the writing of the manuscript, paying special attention to possible orthographic mistakes, and the review of the English writing".

Answer: A professional English editor has gone over the manuscript and corrected all such mistakes.

  1. Comment: Abstract. "I think it is not necessary to indicate that promotion predicts mental well-being, since previously it is written that it mediates the relationship between I-position and mental well-being, and this automatically implies that it predicts mental well-being. Perhaps, it is a good option to change the order of the findings. I think it is better, normally, to explain the relationship between variables first, then the prediction between them, and finally the mediation".

Answer: Thanks to your observation, we now display the results in a different order, presenting the predictions before the mediation. Additional correlations are presented in the Results section.

  1. Comment: Introduction, page 3. Self-regulation subsection. "At the ending sentence, it is said that coping strategies are related to aggressiveness, antisocial behaviours… but although I understand that self-regulation has two dimensions which may are two different styles of dealing with their goals and values, I think it would be better to make this more explicit, that these self-regulation styles may be coping strategies, or perhaps, to indicate that a worse orientation to self-regulation would be linked to the variables explained (aggressiveness, antisocial behaviours…).

Answer: The sentence has been corrected accordingly.

  1. Comment: Introduction, page 3. "Regarding the last paragraph, which starts as “In sum…”, I think it would be better to include this in the next subsection about goals and hypotheses. Also in this paragraph, it is said that DoS is important at a young age. However, the study is not oriented to this goal. I was confused when I saw the sample, since according to this, I was expecting a young sample. I would recommend clarifying this."

Answer: In light of this comment, the paragraph in question has been moved to the next subsection on goals and hypotheses. We also deleted “at a young age” to improve the clarity of the sentence.

  1. Comment: Methods. Instruments subsection. "Cronbach’s alpha coefficients, or reliability coefficient in general, are about the scores of the test, and not about the test itself. Therefore, instead indicating that scale has a good reliability, authors should state that item scores showed a good reliability, or maybe better, just that Cronbach’s alpha coefficient indicated a good reliability."

Answer: We have corrected the wording of the of the scales’ reliabilities.

  1. Comment: Methods. Data analysis subsection. "Authors state that Pearson correlation analysis are conducted for continuous variables, but neither gender nor education are continuous variables. Additionally, I would recommend instead these analyses and ANOVA, a correlation analysis only between the variables of the study, and then a multiple regression analysis, including well-being as the dependent variable, sociodemographic variables at a first step as control variables, and finally the variables of the study (DoS, self-regulation, and anxiety) which were significant at correlation analysis. I think thit would facilitate the analysis and would provide significant information regarding the relationship of all the variables with well-being, controlling the others, in order to ascertain those variables which should be finally included in the mediation model."

Answer: In keeping with your comment, we have made corrections to this subsection (marked with Tracked Changes).

  1. Comment: Results. Mediation model. I think the findings are not explained in the correct order. A mediation analysis is kind of a regression analysis, so you are conducting several regression analyses. First, I-position on prevention, I-position on promotion, I-position on anxiety, I-position on well-being. And second, prevention on well-being, promotion on well-being, and anxiety on well-being. Later the same for emotional cutoff. According to this, if it is previously indicated that a variable mediates between other two variables, this implies that the mediating variable predicts the outcome variable: “Promotion mediated the relationship between I-position and mental well-being” and later it is written “Finally, promotion positively predicted mental well-being”. I would recommend rewriting this subsection, perhaps indicating the relationships between variables first, and finally the ones which had a mediating role. In addition, according to the goals and hypotheses, I understood that anxiety was going to be tested as a mediator, not only between DoS and well-being, but also between self-regulation and well-being. I do not know if this was correct but there was no effect, or whether it never was tested, but I would recommend clarifying this and make the appropriate corrections. Finally, it is also written that “In addition, modification indices indicate a negative direct path between emotional cutoff and mental well-being”. Why is this not indicated for I-position as well? Both have a direct path to well-being, and it would be not necessary to specify this, since these variables are predictors, and well-being the outcome variable.

Answer: These corrections have been made in the Results section. Regarding the relationship between self-regulation and anxiety: we did not assume that one predicts the other, as there is no theoretical or empirical rationale for this. Therefore, it was hypothesized that these two variables will mediate the relationships between DoS and mental well-being.

  1. Comment: Results. I recommend using the same names along all the manuscript. If the authors decided to use “prevention” in the diagram, and along all the manuscript to refer to one of the variables involved in the study, at table 5 this should be the same, instead using “avoidance”.

Answer: Of course you are right. We have replaced “avoidance” with “prevention.”

Again, thanks so much for your feedback. 

Reviewer 2 Report

Overall, the subject matter of this paper is modern, interesting, and useful to the general reader. The authors wrote it concisely, to the point, and well supported by literature.  However, I have some observations issue with manuscript:

- Does this research have an application for research ethics certification in the organization? or have guidelines

- Research tools used, Have permission from the owner been obtained?

- Figure 1 lacks clarity. Please improve thoroughly. and suitable for publication in journals

- Discuss research results well from the writing of the report, however, the recommendations are still unclear. Please suggest suitable for this event. To make us see that you have an interesting way to analyze.

- Double check references

Author Response

Dear reviewer,

We are very grateful for your important comments that undoubtedly contributed to the quality of the manuscript. Below are the corrections we have made:

Reviewer 2

  1. Comments: "Does this research have an application for research ethics certification in the organization? or have guidelines".
  2. "Research tools used, Have permission from the owner been obtained?"

Answers: Details regarding the institution’s approval have been added to the article. Below is the consent form sent to participants:

Greetings,

The following questionnaire contains questions about dealing with different situations. We would be pleased if you could answer the questions. There are no right or wrong answers – only your opinion is important to us. Your participation in the study is of your own free will and you may stop at any time. Participation in the study is expected to take about 20 minutes. We guarantee that all data will be anonymous and discreet and that no use will be made of your answers except for the purpose of this research. No inconveniences are anticipated from study participation. If you have any questions, please contact the researcher by email - orap@yvc.ac.il By going to the next page of the questionnaire, you indicate your consent to participate in the study.

  1. Comment: "Figure 1 lacks clarity. Please improve thoroughly. and suitable for publication in journals"

Answer: We rechecked the figure and added a more accurate caption.

  1. Comment: "Discuss research results well from the writing of the report, however, the recommendations are still unclear. Please suggest suitable for this event. To make us see that you have an interesting way to analyze."

Answer: We have rewritten the Contributions and Clinical Implications section accordingly.

  1. Comment: "Double check references"

Answer: References have been double checked.

Again, thanks so much for your feedback.